# How Ricin Damages the Ribosome

**DOI:** 10.3390/toxins11050241

**Published:** 2019-04-27

**Authors:** Przemysław Grela, Monika Szajwaj, Patrycja Horbowicz-Drożdżal, Marek Tchórzewski

**Affiliations:** Department of Molecular Biology, Maria Curie-Skłodowska University, Akademicka 19, 20-033 Lublin, Poland; monika.szajwaj@poczta.umcs.lublin.pl (M.S.); patrycja.horbowicz@gmail.com (P.H.-D.); maro@hektor.umcs.lublin.pl (M.T.)

**Keywords:** ribosome-inactivating protein, ricin, ribosome, sarcin-ricin loop (SRL), GTPase associated center (GAC), P-proteins, translation

## Abstract

Ricin belongs to the group of ribosome-inactivating proteins (RIPs), i.e., toxins that have evolved to provide particular species with an advantage over other competitors in nature. Ricin possesses RNA N-glycosidase activity enabling the toxin to eliminate a single adenine base from the sarcin-ricin RNA loop (SRL), which is a highly conserved structure present on the large ribosomal subunit in all species from the three domains of life. The SRL belongs to the GTPase associated center (GAC), i.e., a ribosomal element involved in conferring unidirectional trajectory for the translational apparatus at the expense of GTP hydrolysis by translational GTPases (trGTPases). The SRL represents a critical element in the GAC, being the main triggering factor of GTP hydrolysis by trGTPases. Enzymatic removal of a single adenine base at the tip of SRL by ricin blocks GTP hydrolysis and, at the same time, impedes functioning of the translational machinery. Here, we discuss the consequences of SRL depurination by ricin for ribosomal performance, with emphasis on the mechanistic model overview of the SRL *modus operandi*.

## 1. Introduction

Toxic proteins are naturally present in a wide variety of species [1]. It is thought that they have evolved to play a specific role in defense against animals, pathogens, and various insects, giving advantage in a particular niche [2,3]. One of the most toxic proteins known in nature are plant toxins from the class of ribosome-inactivating proteins. The mechanism of toxicity of plant toxins is of great interest because they are present in human foods [4,5] and used in ethnomedicine [6] and cosmetics [7]. Currently, they have attracted attention in broad biotechnological applications [8,9]. Importantly, they also pose a significant threat, as numerous plant toxins, such as ricin, have been recognized as a biological weapon [10] and are recently considered as an agent of bioterrorism [11]. Ricin is one of the most common and potent lethal biological molecules known [9]. Ricin and related proteins display one common feature, i.e. the ability to inhibit ribosomes, and this particular feature laid the foundation for the general name for all these proteins: “ribosome-inactivating proteins” (RIPs) [12]. In general, all RIPs have been classified as single chain (type 1) and double chain (type 2) proteins [13]. Type 1 RIPs consist of an enzymatically active chain (A) and include for instance: pokeweed antiviral protein (PAP), trichosanthin, saporin, gelonin, and luffin. Ricin, abrin, and Shiga toxin are classified as type 2 RIPs, where the enzymatically active A-chain is disulfide-linked to a B-chain, which provides an active chain with higher ability to enter cells. Thus, type 2 RIPs are considerably more toxic than type 1 RIPs [13]. Also, a third class of RIPs, termed type 3, has been recognized. It includes only a few members, with the prominent example of jasmonate induced protein (JIP60) [14] and maize b-32 protein, which requires proteolytic process to become active [15]. In 2010, a new nomenclature has been proposed for RIPs, in which they are termed A (type 1), AB (type 2), AC (type 3, similar to JIP60), and AD (Type 3, similar to b32) [16]. The current model of the catalytic activity of ricin and other RIPs assumes that, using the N-glycosidase activity, the toxin removes a single adenine base from the ribosomal RNA (rRNA) located at the heart of the ribosomal GTPase associated center (GAC) recognized as a central ribosomal element responsible for fueling of the translational machinery [17]. The prime target of ricin is a structural element of rRNA and is called the sarcin–ricin loop (SRL) because this element is targeted by another ribo-toxin, α-sarcin; yet in this case, the SRL undergoes endonucleolytic cleavage [18]. The ricin-dependent depurination of the SRL exerts a deleterious effect, blocking the ribosome action and thereby hampering protein synthesis. It should be stressed that despite the high homology of the SRL within the three domains of life, mainly eukaryotic ribosomes undergo efficient depurination, however, several type I RIPs were shown to depurinate prokaryotic ribosomes as well [19]. The extraordinary specificity of the RIP action is based on their interaction with unique eukaryotic proteins; in the case of ricin, the ribosomal P-proteins are regarded as a main docking part [20,21,22] while the L3 protein represents the main landing platform for the PAP [23,24]. However, regardless of the docking element, the eukaryotic ribosomal proteins are considered as guide molecules for RIPs, directing and activating the toxin toward its catalytic target - the SRL. The toxicity of ricin was mainly associated with inhibition of protein synthesis, but it should be stressed that there is no clear link between depurination of SRL, ribosome hampering and the toxic effect in the cell. The currently available information concerning ricin toxicity suggests that the toxic effect on cell metabolism is multifactorial and involves induction of numerous pathways that lead to cell death; however, molecular details elucidating the effect of ricin on cell metabolism are still elusive [25].

The numerous biochemical approaches [26,27,28,29] and especially current structural analyses have provided deep insight into the role of SRL in ribosome performance and cast light on the molecular consequences of its depurination-induced damage. Here, we are presenting the current understanding of the structure and function of SRL during the translational cycle, with particular focus on the molecular consequences of ricin-dependent adenine base removal on ribosome performance.

## 2. Ribosome Structure as A Prime Target for Ricin

### 2.1. Ribosome and Its Active Sites

Protein biosynthesis, being a critical biological pathway, provides a suitable target for many toxins or natural inhibitors with the ribosome as the main objective [30,31]. The ribosome is one of the most conserved and sophisticated macromolecular machines of the cell in all domains of life [32]. It is composed of a small and a large subunit, which together form a fully functional ribosome. Ribosomal subunits are composed of ribosomal RNA (rRNA) and a large number of ribosomal proteins, but it is the rRNA that plays the most critical functional role, defining the ribosome as a ribozyme [33]. The ribosome can be regarded as a scaffolding platform for many molecules, e.g., mRNA, aminoacyl-tRNAs (aa-tRNA), and protein factors, which together form the translational machinery converting genetic information into functional proteins [34]. The ribosome, being the central element of this machinery, carries out its task through several functional elements. The small subunit decodes the genetic information delivered by mRNA, whereas the large subunit hosts the catalytic peptidyl transferase center (PTC), where amino acids delivered by aa-tRNAs are linked into polypeptides [32]. Additionally, three tRNA binding sites can be recognized on the ribosome: aminoacyl (A), peptidyl (P), and exit (E) sites. The A site accommodates the incoming aa-tRNA, the P site binds the peptidyl-tRNA thus carrying the nascent polypeptide chain, and the E site binds deacylated tRNA before it dissociates from the ribosome. On the large ribosomal subunit there is also the GTPase associated center (GAC) which belongs to the ribosomal elements responsible for administrating the continuous motions of the ribosome along the translational cycle.

#### 2.1.1. The GTPase Associated Center

The translation process to proceed efficiently and to comply with the metabolic needs of the cell requires many protein factors, which sequentially guide the ribosome through the protein synthesis cycle. The most critical group of factors are proteins that bind and hydrolyze GTP, called translational GTPases (trGTPases), which confer the unidirectional trajectory of the translational machinery at the expense of energy released from GTP hydrolysis [17]. The landing platform for all trGTPases is situated on the large ribosomal subunit and is called the GTPase associated center (GAC) (Figure 1) [35].

The GAC is responsible for recruitment of trGTPases and stimulation of factor-dependent GTP hydrolysis [36]. The center consists of two main functional elements: a conserved fragment of rRNA called the sarcin–ricin loop (SRL) and the ribosomal stalk, composed exclusively of ribosomal proteins [37]. Both elements are critical for activation of trGTPases [28,38], and mutual cooperativity of the SRL and the stalk elements has been shown to be pivotal in stimulation of GTP hydrolysis by trGTPases [39,40,41]. 

##### The Sarcin–Ricin Loop

The sarcin–ricin loop is one of the most conserved rRNA regions of the ribosome, which underlines its importance in ribosome function. It is located in helix 95, in domain VI of 23S/25S/28S rRNA (nucleotides 2646-2674 in *E. coli*, 3012-3042 in yeast) (Figure 2A).

Unlike other rRNA regions, which form compact structures coordinated by rRNA–rRNA or rRNA-protein interactions, the SRL exists on the ribosome as an autonomous unit [43,44,45] and is exposed to the solvent [46]. This unique feature is critical for its accessibility for external factors like trGTPases. From the structural point of view, the SRL has the conformation of a distorted hairpin [47]. It consists of several well-organized elements (Figure 2B) [48]: the stem, the flexible region, the G-bulged cross-strand stack, and the GAGA loop [45] with the key adenine base (A_2660_/A_3027_—*E. coli/S.cerevisiae* numbering), which is a target for ricin activity. The SRL stem structure is formed mainly by classical Watson–Crick base pairs, whereas the rest of the structure is stabilized mainly by π-stacking interactions. The critical element, the GAGA tetra-loop, forms a compact well-organized structure [43]. The spatial organization of the loop structure is determined by non-canonical interactions between base pairs, allowing the carbohydrate-phosphate backbone of RNA to form the atypical spatial form, which is recognized by translational factors [43]. The critical bases A_2660_, G_2661_, and A_2662_ in the GAGA loop (*E. coli* numbering) associate with each other via non-canonical π-stacking interactions that stabilize the loop structure [42,43]. The crucial A_2660_, located at the top of the hairpin structure, is stabilized via the stacking interactions with G_2661_ (Figure 2B). Importantly, A_2660_ is fully exposed and does not form any hydrogen bonds with other bases of the loop, making it easily accessible to external factors like ricin [49]. Biochemical studies on prokaryotic and eukaryotic ribosomes have shown that the SRL represents a critical element responsible for the interaction and stimulation of all trGTPases activity [50,51]. Additionally, it has been early recognized that this structure represents the main target for numerous toxins [50,52,53,54]. Recent structural analyses have brought detailed insight into the intricate interplay between the SRL and trGTPases and at the same time cast light on the molecular aspects of ricin toxicity [28,29,40,55]. In general, all trGTPases interact with the SRL via the GTP-binding domain (G-domain) comprising the active site of the factor, responsible for GTP hydrolysis [17]. It should be stressed that the structure of the G-domain is evolutionarily conserved among all trGTPases, and biochemical and structural investigations support the notion that the mechanism of activation of GTP hydrolysis by the ribosome is universally conserved [17,56]. The trGTPases convert chemical energy into mechanical forces at the expense of GTP hydrolysis, and this drives the ribosome through the translational cycle. The key role in catalysis is played by an invariant histidine residue (His_84_ in EF-Tu, His_87_ in EF-G, or His_61_ in SelB, or His_108_ in eEF2) [56,57,58,59,60]. This histidine may adopt two conformations: an inactive “flipped-out” state (pointing away from the γ-phosphate of GTP) and an active flipped-in state (reaching towards the γ -phosphate) (Figure 3). After joining the factor to the ribosome, the SRL is “inserted” into the catalytic center of the G-domain, and the phosphate moiety of A_2662_ coordinates, by means of electrostatic interactions, the catalytic histidine positioning to the active “flipped-in” position towards the γ-phosphate of GTP [28,38,61]. The positively charged histidine points towards the water molecule aimed at the nucleophilic attack on GTP γ-phosphate [29,38,58,62]. Additionally, the phosphate of A_2662_ coordinates a Mg^2+^ ion, important in positioning of the Asp residue (Asp_21_ in EF-Tu, Asp_22_ in EF-G, Asp_10_ in SelB), which is also crucial for GTP hydrolysis (Figure 3) [28,58].

Recently, A_2662_ and G_2661_ within the tetraloop structure were distinguished as critical elements, directly involved in stimulation of the GTP hydrolysis process [28,38]. Interestingly, α-sarcin cleaves the bond between eukaryotic nucleotide equivalents to *E. coli* G_2661_ and A_2662_, which results in ribosome inactivation [63,64]. The A_2660_ base, which is cleaved-off by ricin, plays a distinct but weighty function, which could be named as the “power behind the throne” title role. As shown based on the structure of the EF-G–ribosome complex in a pre-translocation state, an intricate network of hydrogen-based interactions involving the G_2661_ and A_2660_ of the SRL, EF-G (Glu_456_, Arg_660_, Ser_661_ and Gln_664_), and ribosomal L6 (Lys_175_) is formed in the immediate vicinity of the GTPase active site, with A_2660_ being the central element of the network [40]. Thus, depurination of A_2660_ may prevent the surrounding elements from adopting the active conformation, which is required to bind the metal ions necessary to stabilize Asp_22_ and neighboring regions of EF-G in the activated form [55,65]. On the other hand, A_2660,_ together with G_2661,_ plays a crucial role in opening of a so-called hydrophobic gate, which prevents the invariant His residue in the free trGTPase from achieving an active state and spontaneous GTP hydrolysis. In the complex of the EF-G/ribosome, the bases A_2660_ and G_2661_ interact with His_20_ of EF-G, which in turn interacts with Ile_21,_ forming a hydrophobic gate with Ile_63_. These interactions contribute to its opening and repositioning the His_87_ into its active position [40]. The structural analyses of the A_2660_ role are supported by biochemical insight. It has been shown that lack of the single exocyclic N6 amino group at position 2660 within rRNA inhibited GTP hydrolysis on the EF-G/ribosome complex. Importantly, the introduction of different exocyclic groups with dissimilar chemical groups, such as inosine, dimethyladenosine, or even 6-methylpurine, restored the GTP hydrolysis activity of EF-G. The experimental biochemical data indicate that the critical favorable chemical feature is related to electron configuration that allows participation in the aromatic π-electron interaction system of the purine, which in turn facilitates the π-stacking effect [66]. It should be underlined that despite the vast number of data collected from only the bacterial model, the amino acid sequence, called PGH motif (with the invariant His residue) is universally conserved and present in EF-Tu, EF-G, IF2, and RF3, as well as archeal and eukaryotic trGTPases [67]. What is more, the highly conserved A_2660_ base moiety (A_3027_ - yeast, A_4324_ - rat, A_4605_ - human) within the tetra-loop of SRL [56] represents the most crucial base which contributes to a cooperative interaction network, which stabilizes the active state of trGTPases, promoting GTP hydrolysis.

##### The Ribosomal Stalk

The ribosomal stalk represents a vital element within the ribosomal GTPase associated center. Stalk structure is composed of two distinct parts - the base of the stalk and its lateral elements [68]. The stalk base is constituted by conserved ribosomal proteins uL11 and uL10, which anchor the stalk to the rRNA [69,70]. The lateral part of the stalk has multimeric architecture and is built of multi-dimeric protein elements, which are unique for bacteria and eukaryotes. In bacteria, the bL12 proteins form a dimer, which is regarded as a basic structural element, and two, three, and even four dimers can form the lateral part anchored to the ribosome through uL10 [35,71]. In eukaryotes, the P1 and P2 proteins form a dimer, and two dimers are linked to the uL10 protein, forming pentameric architecture called the P-stalk, uL10-(P1-P2)_2_ [37,69,70,72,73,74,75]. It should be underlined that the stalk fulfils the same function on the ribosome, irrespectively of the life-domain origin, namely participation in stimulation of GTP hydrolysis [68]; however, the bL12 and P1/P2 proteins are not evolutionarily related and are regarded as analogous proteins [76,77]. The stalk is the only structure on the ribosome composed of multiple proteins. The eukaryotic stalk architecture has a complex nature. It is constituted by two P1/P2 protein dimers; each dimer is built of two domains: an N-terminal globular domain (NTD), responsible for dimerization and anchoring the dimer to uL10, and an unstructured C-terminal domain (CTD), regarded as a functional part interacting with trGTPases [70,73,78]. Both elements are connected through a highly flexible hinge region [73,79]. The most prominent feature of the eukaryotic P1/P2 stalk proteins is the highly conserved element present at the CTD, composed of a stretch of acidic and hydrophobic amino acids (EESEESDDDMGFGLFD) and regarded as the main functional element of the stalk. This element is involved in the interaction with trGTPases and toxins such as ribosome-inactivating proteins (RIPs) [22,80,81,82,83,84]. A unique feature of the eukaryotic stalk is multiplication of CTDs. The conserved CTD is also found on the uL10; therefore, five CTDs are present on the stalk: four coming from two P1/P2 dimers and one from uL10. The phenomenon of CTDs multiplication was functionally coupled with the qualitative aspect of ribosome action related to maintenance of translation accuracy [85]. It was proposed that the multiple CTDs might accelerate interaction with eEF1A, which is regarded as trGTPase with the highest GTP hydrolysis turnover. Interestingly, this feature has been hijacked by RIP toxins, and it has been shown that multiplication of P1/P2 proteins increase the interaction rate of the toxin [86].

### 2.2. Mode of Ricin Interaction with Ribosome

It has been established that ricin inhibits translation through its ability to remove/depurinate a specific adenine base of the universally conserved SRL [87], which is a crucial part of the GAC on the ribosome [27,38,88,89]. The SRL has been found as a primary target for ricin and other RIPs, and the specificity of the interaction with eukaryotic ribosomal proteins plays a critical role in ricin catalytic activity towards SRL. As shown over two decades ago, the efficiency of rRNA depurination in the intact ribosome is much greater than the depurination of isolated 28S rRNA. The *k*_cat_ of ricin against naked rRNA is more than 4 orders of magnitude lower than that of rRNA constituting a part of the ribosome [90,91,92]. Ricin depurinates the naked 23S rRNA from *E. coli* SRL, but not the intact ribosomes from *E. coli* [91], showing at the same time extraordinary specificity towards intact eukaryotic ribosomes [20,21,93], what underlines the role of ribosomal proteins in the process. The same applies to other related RIPs, such as Shiga toxin 1 (Stx1) [93,94], Shiga toxin 2 (Stx2) [95,96], trichosanthin (TCS) [97,98,99], and maize RIP [100], which specifically depurinate the SRL on the eukaryotic ribosome. In the case of ricin, the mechanistic model of molecular recognition of the ribosome assumes a double-step mechanism, involving first slow and nonspecific electrostatic-based interactions with the ribosome and then fast specific interactions based on the ribosomal stalk interplay, leading to its attack on the SRL rRNA [90]. Although the SRL is highly conserved among ribosomes in all species, the P-proteins determine the specificity of ricin and other RIPs toward eukaryotic ribosomes [12,22]. The deletion of stalk P-proteins from ribosomes greatly reduces the depurination activity and cellular sensitivity to ricin, indicating that binding to the P-stalk is a critical step in depurination of the SRL and in the toxicity of ricin [101,102,103]. The structural investigations provide significant insights into the mode of interaction between P-proteins and the ricin or trichosanthin (TCS), which hijacks the translational factor recruitment function of the ribosomal P-stalk to reach its target site on the ribosome [22,84]. Especially, the interaction site of P-proteins with RIPs was mapped to a short conserved 11-mer peptide, SDDDMGFGLFD, present at the CTDs of all P-proteins [98]. This interaction is required for the full activity of ricin and other RIPs, and biochemical analyses confirmed that positively charged residues, especially the cluster of arginines, play a key role [101]. As was shown in a TCS study, the interaction of TCS is primarily mediated by the electrostatic interactions of K_173_, R_174_, and K_177_ in the C-terminal domain of TCS with the conserved DDD residues in the CTDs of P-proteins. However, hydrophobic interactions also play a vital role in stabilization of the bilateral interplay between TCS and the conserved C-terminal peptide of P-proteins [93,99]. The tertiary structure of the catalytic subunit of ricin (RTA) with a short peptide corresponding to the last six conserved residues of the stalk proteins (GFGLFD) showed that the peptide docks into a hydrophobic pocket at the C-terminus of RTA [84,104]. The structural superposition of TCS-P-protein and RTA-P-protein complexes demonstrated that the short C-terminal peptide of P-proteins adopts distinct orientations and slightly different interaction modes with the two different RIPs, suggesting that the flexibility of the CTD facilitates accommodation of different class of RIPs to the ribosome [84,104]. The kinetic studies showed that the P1-P2 heterodimeric conformation of P-proteins in the stalk pentamer represents an optimal binding site for RTA, where individual P-protein CTDs play non-equivalent roles with a pivotal role of P1 CTD [65]. Additionally, previous results obtained using yeast as an experimental system showed that the two dimers, P1A-P2B and P1B-P2A, do not interact equally with RTA [102], suggesting that these dimers may have a different architecture and their CTDs may not be equally accessible to external factors, such as RTA or other RIPs. The high specificity of ricin interaction with the P-stalk is also reflected by the measured dissociation constant, which is in a nanomolar range [65,80,86]. Thus, the kinetic model of RTA interaction with the ribosome and SRL depurination assumes that the toxin initially interacts with the P-protein stalk, and it allows orienting the active site of the toxin toward the SRL, which in turn places it in correct orientation for binding to the target adenine. It is also proposed that the P-stalk binding event allosterically stimulates the catalysis of ribosome depurination by RTA, explaining the extraordinary specificity of the toxin toward eukaryotic ribosomes [101]. 

## 3. Toxic Action of Ricin on The Translational Process

Ricin is composed of two subunits, RTA and RTB, covalently linked through a disulfide bond. In the form of holotoxin, it does not exhibit catalytic activity toward ribosome [105]. When RTA is separated from RTB, the cluster of arginine residues located at the interface domain between RTA and RTB is exposed to the solvent and serves as an interaction platform for P-stalk proteins. The RTA-stalk interaction stimulates the toxin to trigger its enzymatic activity by orienting the active site of RTA (opposite to the arginine interface) toward the SRL [101]. RTA is an RNA N-glycosidase (EC 3.2.2.22) that hydrolyzes the N-glycosidic bond between a specific adenine on the SRL and the sugar backbone [91]. The specificity of rRNA depurination by RTA is determined by the conformation of the topical part of the SRL loop structure, and the GAGA sequence with the prominent key adenine base is recognized as the major element [106,107]. During the catalysis of the SRL depurination process by ricin, the conserved adenine on the tip of the sarcin–ricin loop is inserted between two Tyr residues (in the RTA catalytic center - Tyr_80_ and Tyr_123_) to form π-stacking interactions [108,109]. Additionally, the adenine position is stabilized by hydrogen binding with RTA Gly_121_, Val_81_, Glu_177_, and Arg_180_ residues [110]. It has been shown that two RTA residues, i.e. Glu_177_ and Arg_180_, play a crucial role in the hydrolysis of N-glycosidic bonds by stabilizing the transition state during catalysis of the depurination reaction (Figure 4) [111,112].

As already discussed, the structural stability of the SRL is provided mostly by the π-stacking interaction network [52,113,114]. Removal of the key adenine at the tip of the SRL may destabilize this type of interactions, thereby affecting the SRL structure stability and abolishing an extended interaction network responsible for stabilization of the active state of trGTPases (Figure 4). 

It was already observed in the 1970s that ricin inhibits translation in mammalian cells [115], as confirmed in an in vitro experimental system [53,116,117,118,119]. Early analyses in in vitro protein synthesis systems have shown that the presence of ricin blocks the synthesis of polypeptides, and this was mainly associated with the elongation step of the translational cycle [116,117,120,121,122]. It has also been reported that ricin does not affect the synthesis of peptide bonds [53,116,123,124], but a significant inhibition of GTP hydrolysis was associated with ricin action [53,54,116,124,125,126,127,128,129,130]. Numerous analyses have shown that eEF2, i.e. a factor involved in the translocation event during the elongation cycle, binds less efficiently to ribosomes modified by ricin [54,125,126,128,131]. Additionally, it has also been shown that treatment of ribosomes with ricin decreased the level of GTP hydrolysis by the eEF2 factor [53,116]. Further analyses demonstrated that ricin inhibited translocation, and the effect was dependent on the eEF2 concentration used [123]. Additional evidence for the inhibitory effects of ricin on translocation was provided by applying test with the use of diphtheria toxin, showing that the ribosomes treated with both toxins mainly paused at the beginning of the mRNA [119]. It was also shown that the rate of formation of pre-initiation complexes, i.e. the attachment of the 60S subunit to the pre-initiation complex 40S, was decreased under the influence of ricin [119]. Thus, these analyses have provided evidence that the modification of the 60S subunit by ricin resulted not only in inhibition of elongation at the translocation step, but also in reduction of the initiation rate [119]. In vitro analyses on the bacterial model have confirmed the experiments performed on the eukaryotic system, showing that ribosomes lacking adenine in the topical part of SRL are unable to stimulate GTP hydrolysis by EF-G, which is a bacterial trGTPase homologous to eukaryotic eEF2 [66]. All these experiments laid the foundation for a general notion that depurination of the SRL brings deleterious effects for the translational machinery, linking the toxic effect of ricin with blockage of protein synthesis in the cell. However, in vivo analyses have shown that there is no clear cross-correlation between the ribosome depurination, translation inhibition, and cell death [102,103], leaving the issue of ricin toxicity at the molecular level as an open question. All currently available information concerning ricin toxicity suggests that the toxic effect on cell metabolism has a multifactorial nature, involving induction of numerous pathways leading to cell death [25], but the molecular trigger is still obscure.

## 4. Conclusions

Ricin, in particular its catalytic subunit RTA, targets one of the most important eukaryotic ribosomal catalytic centers—the GAC, which is responsible for conferring unidirectional trajectory for the translation apparatus at the expense of GTP hydrolysis driven by trGTPases. To access the GAC, ricin hijacks the ribosomal translation factor recruitment element, i.e., the P-stalk, to reach the target adenine base in the SRL on the ribosome. The stalk, composed of P-proteins, represents a unique eukaryotic element interacting with RTA, being responsible for the specificity of the toxin toward the eukaryotic ribosome. The RTA interaction with the stalk not only anchors and directs RTA towards SRL but also stimulates depurination of invariant adenine at the tip of the SRL. The SRL represents a critical ribosomal element responsible for triggering GTP hydrolysis by trGTPases. The G_2659_ A_2660_G_2661_A_2662_ (*E. coli* numbering) tetra-loop located on the tip of the SRL plays a key role here. Within this loop, two bases A_2662_ and G_2661_ are critical elements directly involved in stimulation of GTP hydrolysis [28]. On the other hand, A_2660_, i.e., a base that is depurinated by RTA, is situated away from the trGTPase active site, being a center of cooperative interaction network, contributing to stabilization of the active state of trGTPases and promoting GTP hydrolysis. Thus, the A_2660_, lying away from the main catalytic center, but coordinating the structural arrangement of the G domain of trGTPases, could hold the “power behind the throne” role. Therefore, depurination of solvent-exposed A_2660_ impairs the intricate interaction network and destabilizes the active state of trGTPases, which finally blocks GTP hydrolysis and, at the same time, impedes the functioning of the translational machinery. However, the majority of biochemical experiments with in vitro depurinated ribosomes have shown unequivocally that removal of A_2660_ blocks translation, especially at the elongation step of the translational cycle, linking the catalytic activity with ricin toxicity. Such a situation seems to be unusual inside the cell, where most of the toxin molecules are degraded during the ricin trafficking and, as it was estimated, only up to 5% of RTA molecules reach the endoplasmic reticulum [132,133,134]. Thus, ricin must have evolved to be extremely successful in winning the battle to hurt the GAC - the energetic heart of the ribosome; this is especially well demonstrated by the higher affinity towards the ribosomal P-stalk proteins [21,86,102] compared to translational factors [82]. Importantly, in vivo analyses did not find any clear link between the ribosome depurination, translation inhibition, and cell death, indicating that molecular events contributing to the so-called ‘cause and effect’ of the ricin *modus operandi* are still obscure.

## Figures and Tables

**Figure 1 toxins-11-00241-f001:**
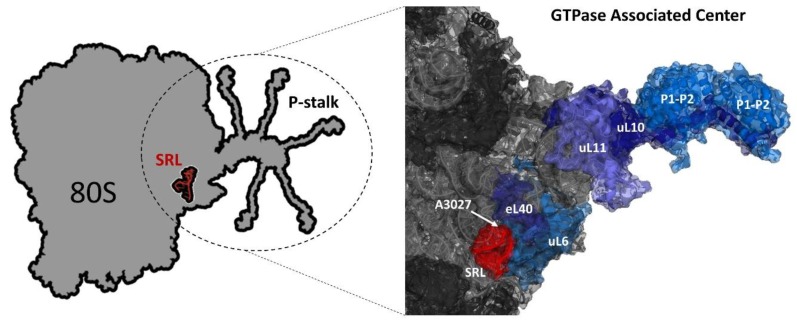
Model of GTPase associated center (GAC). Left panel: scheme of the 80S ribosome with the ribosomal P-stalk shown with extended C-terminal regions and the sarcin–ricin loop (SRL) (red). Right panel: fragment of 60S *S. cerevisiae* ribosome 25S rRNA (PDB code 3U5H) and 60S subunit (PDB code 3U5I); 25S rRNA and ribosomal proteins are indicated as light gray and dark gray colors, respectively. The fitted schematic structure of uL10 protein fragment in complex with the N-terminal domains of P1/P2-proteins (PDB code 3A1Y) from Archaea is depicted as dark blue and marine blue, respectively. The position of the yeast 60S subunit is oriented to show A_3027_ of the SRL and positions of uL11 (slate blue), uL40 (deep blue) and uL6 (sky blue) proteins. Model prepared with PyMol software (The PyMOL Molecular Graphics System, version 1.5.0.4, Schrödinger, LLC, NY, USA).

**Figure 2 toxins-11-00241-f002:**
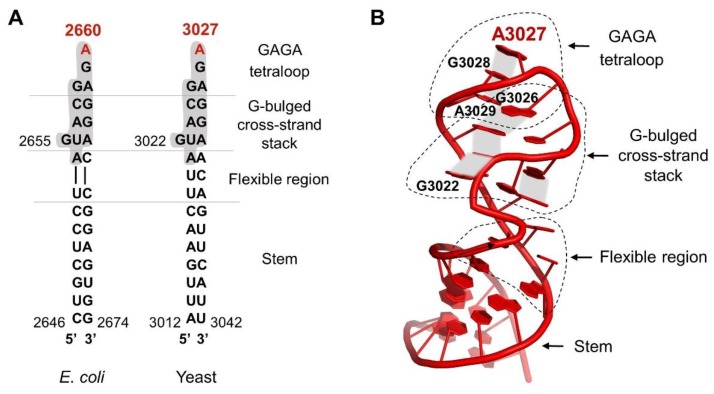
The model of sarcin–ricin loop structure. (**A**) Alignment of highly conserved secondary structures of yeast and *E. coli* SRL. The red color indicates the key adenine hydrolyzed by the ricin. A conserved fragment of 12 nucleotides is marked with a gray color. (**B**) Structure of the *S. cerevisiae* SRL (PDB code 3U5H). The key adenine was marked in red (A_3027_). The individual structural elements of the SRL - the stem, the flexible region, the G-bulged cross-strand stack with the highlighted individual nucleotides and the GAGA loop - are marked with a dotted black line. The gray fields show the non-canonical π-stacking interactions between particular bases. Model prepared by PyMol software, (The PyMOL Molecular Graphics System, version 1.5.0.4, Schrödinger, LLC, NY, USA) based on [42].

**Figure 3 toxins-11-00241-f003:**
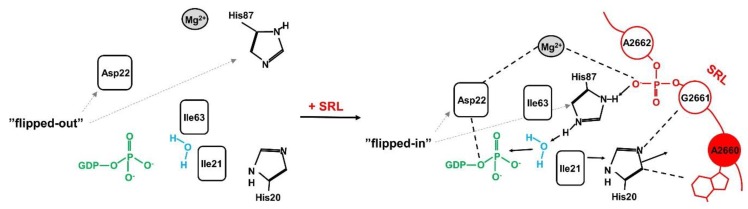
Model of GTP hydrolysis activation with the aid of sarcin–ricin loop (SRL), with the EF-G as trGTPase. Left panel: the organization of the active site in isolated EF-G. Asp_22_ and His_87_ are in “flipped-out” state, pointing away from GTP; the “hydrophobic gate” formed by amino acids Ile_63_ and Ile_21_ prevents His_87_ from adopting the active conformation. Right panel: the reorganization of the active site of EF-G as a result of binding of EF-G to ribosome and inserting the SRL to the G-domain. Base A_2660_ interacts with His_20_, which induces Ile_21_ movement away from GTP and “hydrophobic gate” opening. The phosphate of A_2662_ directs His_87_ and Asp_22_ residues (through Mg^2+^) to “flipped-in” conformation which allows for water molecule activation and GTP hydrolysis (see the details in the text).

**Figure 4 toxins-11-00241-f004:**
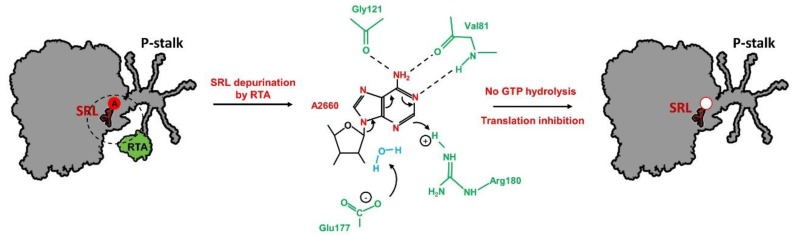
Model of sarcin–ricin loop (SRL) depurination by RTA. Removal of key A_2660_ residue (*E. coli* numbering) impairs the intricate interaction network responsible for stabilization of the active state of trGTPases, resulting in the inhibition of translation process. The base of A_2660_ is bound to the catalytic center of RTA with π-stacking interactions and its position is stabilized by hydrogen bonds with Gly_121_, Val_81_ and Arg_180_ (green). Glu_177_ and Arg_180_ (green) play crucial role in catalysis by transition state stabilization. Arg_180_ residue protonates base of A_2660_ causing delocalization of ring electrons. Glu_177_ polarizes water molecule (blue) and resultant hydroxide ion attacks positive center on ribose which leads to hydrolysis of N-glycosidic bond; prepared based on [110].

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
