# Peer review of "How Ricin Damages the Ribosome"

_toxins, 2019, doi:10.3390/toxins11050241_

Round 1
Reviewer 1 Report
The subject addressed in this review is very interesting. The authors approach the toxic action of Ricin on the ribosome, with particular focus on the structure and function of SRL stem during the translational cycle. The manuscript is written in a very proper and scientific way. It contains all important information and current reports are discussed. Therefore, I strongly recommend publication of this manuscript in Toxins. English language, grammar, punctuation, spelling, and overall style are fine.
Although the actual version of the manuscript is suitable for publication, I list a few minor points that might be considered by the authors in order to improve the paper:
Line 23: Please include the reference: Di Maro et al. “Sequence comparison and phylogenetic analysis by the Maximum Likelihood method of ribosome-inactivating proteins from angiosperms” Biochimica et Biophysica Acta 1760 (2006) 783–792.
Line 25: Please remove reference 3 and cite a more appropriate one, which includes a detailed study of the presence of RIPs in edible plants: Barbieri et al. “Ribosome-inactivating proteins in edible plants and purification and characterization of a new ribosome-inactivating protein
from Cucurbita moschata” . Biochimica et Biophysica Acta 1760 (2006) 783–792.
Line 50: The authors should indicate in this paragraph, that it has been published that several type I RIPs can depurinate prokaryotic ribosomes. Ref: Iglesias et al. “Biological and antipathogenic activities of ribosome-inactivating proteins from Phytolacca dioica L.” Biochimica et Biophysica Acta 1860 (2016) 1256–1264.
Author Response
Response to Reviewer 1 Comments
Point 1:
Line 23: Please include the reference: Di Maro et al. “Sequence comparison and phylogenetic analysis by the Maximum Likelihood method of ribosome-inactivating proteins from angiosperms” Biochimica et Biophysica Acta 1760 (2006) 783–792.
Response 1: According to Reviewer suggestion the reference was included with corrected publisher data: Di Maro, A. et al., Sequence comparison and phylogenetic analysis by the Maximum Likelihood method of ribosome-inactivating proteins from angiosperms. Plant Mol Biol, 2014. 85(6): p. 575-88.
Point 2:
Line 25: Please remove reference 3 and cite a more appropriate one, which includes a detailed study of the presence of RIPs in edible plants: Barbieri et al. “Ribosome-inactivating proteins in edible plants and purification and characterization of a new ribosome-inactivating protein from Cucurbita moschata”. Biochimica et Biophysica Acta 1760 (2006) 783–792.
Response 2: According to Reviewer suggestion the reference was included.
5. Barbieri, L., et al., Ribosome-inactivating proteins in edible plants and purification and characterization of a new ribosome-inactivating protein from Cucurbita moschata. Biochim Biophys Acta, 2006. 1760(5): p. 783-92.
Point 3:
Line 50: The authors should indicate in this paragraph, that it has been published that several type I RIPs can depurinate prokaryotic ribosomes. Ref: Iglesias et al. “Biological and antipathogenic activities of ribosome-inactivating proteins from Phytolacca dioica L.” Biochimica et Biophysica Acta 1860 (2016) 1256–1264.
Response 3: According to Reviewer suggestion the paragraph was changed, and appropriate reference was included:
Was: “It should be stressed that despite the high homology of the SRL within the three domains of life, only eukaryotic ribosomes undergo depurination.”
Changed to: “It should be stressed that despite the high homology of the SRL within the three domains of life, mainly eukaryotic ribosomes undergo efficient depurination, however several type I RIPs were shown to depurinate prokaryotic ribosomes as well [Iglesias, R., et al., Biological and antipathogenic activities of ribosome-inactivating proteins from Phytolacca dioica L. Biochim Biophys Acta, 2016. 1860(6): p. 1256-64.].”

Reviewer 2 Report
The Review “How ricin hurts ribosome and translation” briefly introduces the reader to the major catalytic centers of the ribosome and extensively to the GTPase associated center (GAC), the role of the sarcin-ricin loop (SRL) as GTPase activating element, the universal ribosomal stalk, how ricin finds its sarcin-ricin loop target and what happens to protein synthesis after de-purination of adenine 2660 in the SRL. The review is undoubtedly written by aficionados of the subject area with a very deep interest in both the stalk and the SRL. There is a lot of useful detail regarding GTPase activation and other central points regarding protein synthesis, often with a slant towards structure. There are many ways to write a review on this topic and I am very much in favor publication of this one due to its meticulous detail and, not the least, love for the subject area. I have some suggestions or questions regarding choice and meaning of the language used, mainly concerning deconvolution of convoluted sentences, as follows:
Line 45, suggestion: The prime target of ricin is a structural element of rRNA and is called the sarcin-ricin loop (SRL) because this element is targeted by another ribo-toxin, ……
Line 48, suggestion: blocking the ribosome action and thereby hampering protein synthesis.
Or, do the authors want to keep these functions separate, in line with subsequent discussions, in which case a clarification is warranted?
Line 69, suggestion: …pathway, provides suitable…
Line 72, suggestion; It is composed of a small and a large subunit, …..(unequal redundant or tantalizing)
Line 113, suggestion: ,,,underlines (reflects) its importance…
Line 142, suggestion, e.g.: Biochemical studies on….
Line 214, discussion point: A unique feature is…Two questions involved i.e. several identical proteins and an ensemble of proteins which are both of same and different types. This could preferentially be clarified further!
Line 240: the meaning which refers to eukaryote vs bacterial ribosomes could preferentially be written more explicitly for clarity!
Line 311: maybe present tense here, i.e that ricin inhibits translation….
Author Response
Response to Reviewer 2 Comments
Point 1:
Line 45, suggestion: The prime target of ricin is a structural element of rRNA and is called the sarcin-ricin loop (SRL) because this element is targeted by another ribo-toxin, ……
Response 1: According to Reviewer suggestion the paragraph was changed.
Was: “The prime target of ricin, i.e. the rRNA structure is called the sarcin-ricin-loop (SRL) because this structure is also targeted by another ribotoxin, α-sarcin; yet in this case, the SRL undergoes endonucleolytic cleavage.”
Changed to: “The prime target of ricin is a structural element of rRNA and is called the sarcin-ricin loop (SRL) because this element is targeted by another ribo-toxin, α-sarcin; yet in this case, the SRL undergoes endonucleolytic cleavage.”
Point 2:
Line 48, suggestion: blocking the ribosome action and thereby hampering protein synthesis.
Or, do the authors want to keep these functions separate, in line with subsequent discussions, in which case a clarification is warranted?
Response 2: According to Reviewer suggestion the paragraph was changed.
Was: “The ricin-dependent depurination of the SRL exerts a deleterious effect, blocking the ribosome action and hampering protein synthesis at the same time.”
Changed to: “The ricin-dependent depurination of the SRL exerts a deleterious effect, blocking the ribosome action and thereby hampering protein synthesis.”
Point 3:
Line 69, suggestion: …pathway, provides suitable…
Response 3: According to Reviewer suggestion the paragraph was changed.
Was: “Protein biosynthesis, being a critical biological pathway, represents a very suitable target for many toxins or natural inhibitors with the ribosome as the main objective.”
Changed to: “Protein biosynthesis, being a critical biological pathway, provides suitable target for many toxins or natural inhibitors with the ribosome as the main objective.”
Point 4:
Line 72, suggestion; It is composed of a small and a large subunit, …..(unequal redundant or tantalizing)
Response 4: According to Reviewer suggestion the paragraph was changed.
Was: “It is composed of a two unequal small and a large subunits, which together form a fully functional ribosome.”
Changed to: “It is composed of a small and a large subunit, which together form a fully functional ribosome.”
Point 5:
Line 113, suggestion: ,,,underlines (reflects) its importance…
Response 5: According to Reviewer suggestion the paragraph was changed.
Was: “The sarcin-ricin loop is one of the most conserved rRNA regions of the ribosome, which underlies its importance in ribosome function.”
Changed to: “The sarcin-ricin loop is one of the most conserved rRNA regions of the ribosome, which underlines its importance in ribosome function.
Point 6:
Line 142, suggestion, e.g.: Biochemical studies on….
Response 6: According to Reviewer suggestion the paragraph was changed.
Was: “A biochemical studies on prokaryotic and eukaryotic ribosomes have shown that the SRL represents a critical element responsible for the interaction and stimulation of all trGTPases activity.”
Changed to: “Biochemical studies on prokaryotic and eukaryotic ribosomes have shown that the SRL represents a critical element responsible for the interaction and stimulation of all trGTPases activity.”
Point 7:
Line 214, discussion point: A unique feature is…Two questions involved i.e. several identical proteins and an ensemble of proteins which are both of same and different types. This could preferentially be clarified further!
Response 7: According to Reviewer suggestion the paragraph was changed.
Was: “A unique feature of the stalk is the fact that this is the only ribosomal structure composed of multiple proteins.”
Changed to: “The stalk is the only structure on the ribosome composed of multiple proteins.”
Point 8:
Line 240: the meaning which refers to eukaryote vs bacterial ribosomes could preferentially be written more explicitly for clarity!
Response 8: According to Reviewer suggestion the paragraph was changed.
Was: “Ricin depurinates the naked 23S rRNA from E. coli SRL, but not the intact ribosomes from E. coli, showing at the same time extraordinary specificity towards eukaryotic ribosomes.”
Changed to: “Ricin depurinates the naked 23S rRNA from E. coli SRL, but not the intact ribosomes from E. coli, showing at the same time extraordinary specificity towards intact eukaryotic ribosomes, what underlines the role of ribosomal proteins in the process.”
Point 9:
Line 311: maybe present tense here, i.e that ricin inhibits translation….
Response 9: According to Reviewer suggestion the paragraph was changed.
Was: “It was already observed in the 1970s that ricin inhibited translation in mammalian cells, as confirmed in an in vitro experimental system.”
Changed to: “It was already observed in the 1970s that ricin inhibits translation in mammalian cells, as confirmed in an in vitro experimental system.”
